# Integrative Analysis of Single-Cell and Bulk Sequencing Data Depicting the Expression and Function of P2ry12 in Microglia Post Ischemia–Reperfusion Injury

**DOI:** 10.3390/ijms24076772

**Published:** 2023-04-05

**Authors:** Chenglong Wang, Li Peng, Yuan Wang, Ying Xue, Tianyi Chen, Yanyan Ji, Yishan Li, Yong Zhao, Shanshan Yu

**Affiliations:** 1Department of Pathology, College of Basic Medicine, Chongqing Medical University, Chongqing 400016, China; 2Molecular Medicine Diagnostic and Testing Center, Chongqing Medical University, Chongqing 400016, China; 3Department of Pathology, The First Affiliated Hospital of Chongqing Medical University, Chongqing 400016, China

**Keywords:** *P2ry12*, microglia, ischemia–reperfusion injury, bioinformatics

## Abstract

*P2ry12* is a microglial marker gene. Recently, increasing evidence has demonstrated that its expression levels can vary in response to different CNS disorders and can affect microglial functions, such as polarization, plasticity, and migration. However, the expression and function of *P2ry12* in microglia during ischemia–reperfusion injury (IRI) remain unclear. Here, we developed a computational method to obtain microglia-specific *P2ry12* genes (MSPGs) using sequencing data associated with IRI. We evaluated the change in comprehensive expression levels of MSPGs during IRI and compared it to the expression of *P2ry12* to determine similarity. Subsequently, the MSPGs were used to explore the *P2ry12* functions in microglia through bioinformatics. Moreover, several animal experiments were also conducted to confirm the reliability of the results. The expression of *P2ry12* was observed to decrease gradually within 24 h post injury. In response, microglia with reduced *P2ry12* expression showed an increase in the expression of one receptor-encoding gene (*Flt1*) and three ligand-encoding genes (*Nampt*, *Igf1*, and *Cxcl2*). Furthermore, double-labeling immunofluorescence staining revealed that inhibition of P2ry12 blocked microglial migration towards vessels during IRI. Overall, we employ a combined computational and experimental approach to successfully explore *P2ry12* expression and function in microglia during IRI.

## 1. Introduction

Stroke is a major health concern that results in death and disability around the world. It is divided into two categories: ischemic and hemorrhagic, with ischemic strokes being the most prevalent, accounting for almost 80% of all stroke cases [1]. Reperfusion, the process of restoring blood flow to an ischemic area, is a common treatment goal for ischemic strokes. However, some studies have clearly demonstrated a deleterious effect of reperfusion injury following ischemic stroke using transient middle cerebral artery occlusion (tMCAO) in rodents and radiological examination in human patients with ischemia–reperfusion injury (IRI) [2,3]. Microglia, which make up about 10% of brain cells [4], are the primary innate immune cells in the brain and play important roles in various neurological disorders, including stroke, Alzheimer’s disease, and Parkinson’s disease. Several studies have also demonstrated microglia’s key role in IRI [5,6,7]. Despite this, the role of microglia in IRI is still unclear and can be both beneficial and harmful [8,9]. Thus, understanding how microglial functions shift during IRI could lead to the discovery of therapeutic targets for alleviating IRI.

The *P2ry12* is a member of the P2 purinergic receptor family and is predominantly expressed in platelets and microglia. Despite being a microglial marker gene, *P2ry12* expression levels fluctuate. The expression levels of *P2ry12* in microglia can vary under different physiological and pathological conditions. For instance, under normal conditions, the expression of *P2ry12* in human microglia decreases with age [10], while in *mice*, it first increases and then decreases with age [11]. In pathological conditions, the expression of P2ry12 in microglia varies greatly depending on the type of CNS disorder. For example, the expression of P2ry12 in microglia is decreased in Alzheimer’s disease and multiple sclerosis [12,13,14] and increased in ischemic stroke [15]. However, the expression of *P2ry12* in microglia during IRI is not yet understood. It is also worth noting that *P2ry12* expression has been found in a variety of cells [16,17,18,19], including vascular smooth muscle cells, endothelial cells, dendritic cells, macrophages, and leukocytes. As a result, it is uncertain whether *P2ry12* expression in the cerebral cortex can serve as a substitute for microglial *P2ry12* expression.

Several studies have demonstrated the involvement of P2ry12 in the regulation of microglial function in the pathophysiology of various CNS disorders. For example, deactivation of P2ry12 can cause a change in microglial shape from a ramified to an amoeboid morphology and activate M2 polarization, while activation of P2ry12 can promote microglial migration to the injury region [20,21,22]. However, the function of P2ry12 in microglial cells during IRI remains unclear, and the use of knockout techniques to study the gene’s function appears to be not possible due to the crucial role of P2ry12 in platelet activation and blood clotting, which could lead to a lethal hemorrhagic stroke in IRI-associated models [23]. Thus, the role of P2ry12 in microglia during IRI remains unclear and warrants further research.

In this study, we aimed to shed light on the expression and function of the *P2ry12* gene in microglia during IRI. To achieve this, we developed a computational method based on sequencing data and conducted several experiments in vivo. The recent advancements in single-cell RNA sequencing (scRNA-seq) techniques and analytical methods have opened new avenues for researchers to gain a deeper understanding of biological systems by classifying, characterizing, and distinguishing cells at the transcriptome level [24]. By utilizing scRNA-seq and bulk sequencing (bulk-seq) data of the cerebral cortex in *mice* with IRI, we were able to identify 88 microglia-specific *P2ry12* genes (MSPGs). Following a thorough assessment of the equivalency of *P2ry12* and MSPGs, we used MSPGs to investigate the expression and function of *P2ry12* in cortical microglia during IRI. Our findings revealed that the expression of *P2ry12* in microglia gradually declines within 24 h post injury and that *P2ry12* can modulate the migration of microglia to the vessels during IRI. Moreover, this study provides novel insights into the role of *P2ry12* in microglial function during IRI and highlights the potential of computational methods based on sequencing data to further our understanding of this complex biological system.

## 2. Results

### 2.1. Demographic Characteristics of scRNA-seq Datasets Associated with IRI

The scRNA-seq data GSE167593 contained one ipsilateral cortex sample from a tMCAO *mouse* and one normal cortex sample from a sham *mouse*, and GSE174574 contained three ipsilateral cortex samples and three normal cortex samples. The data underwent quality control and technical dropout imputation, and t-SNE analysis based on the Harmony-corrected latent space was performed to show an effective integration performance between the tMCAO and sham samples (Figure 1a,b). Our optimized annotating method was used to identify nine cell populations in GSE167593, including lymphocytes, oligodendrocytes, microglia, neural progenitor cells, monocytes/macrophages, neurons, astrocytes, granulocytes, and ependymal cells (Figure 1c and Appendix Aa). In GSE174574, ten cell populations were identified, including endothelial cells, monocytes/macrophages, ependymal cells, microglia, oligodendrocytes, astrocytes, perivascular cells, lymphocytes, granulocytes, and neural progenitor cells (Figure 1d and Appendix Ab).

To assess the expression of *P2ry12* across all populations, the projection of the imputed *P2ry12* expression value on t-SNE was generated. A consistent pattern of high *P2ry12* expression only in the microglia population was observed in the two data (Figure 1e,f). The finding was further supported by the projection of the pre-imputed and imputed *P2ry12* expression value in different groups (Appendix A). Moreover, to assess the effect of imputed expression profiles on the *P2ry12* expression value, a comparative analysis between pre-imputed and imputed expression profiles was conducted. In GSE167593, a zero-inflated distribution of the *P2ry12* expression value, regardless of the populations and groups, was observed in the two expression profiles (Figure 1g,h). The dropout ratio of *P2ry12* in microglia showed a decrease from 29.3% in the pre-imputed value to 11.8% in the imputed value, while in monocytes/macrophages, the decrease was from 71.1% to 65.6% in the tMCAO group and from 73.2% to 66.9% in the sham group. The dropout ratio for the rest of the populations remained above 89% (Appendix A). In the tMCAO group of GSE174574, the zero-inflated distribution of the *P2ry12* expression value was observed in both expression profiles, regardless of the population (Figure 1i). The dropout ratio of *P2ry12* in microglia decreased from 37.6% in the pre-imputed value to 15.1% in the imputed value, while in monocytes/macrophages, the decrease was from 85.0% to 75.9%. The dropout ratio for neural progenitor cells decreased from 83.0% to 70.2%, while for the rest of the populations, the dropout ratio remained above 85% (Appendix A). In the sham group of GSE174574, the pre-imputed *P2ry12* expression value showed a zero-inflated distribution across all populations, except for microglia (1.5%), but the imputed expression value did not (Figure 1j). The dropout ratio across all populations in the pre-imputed value was between 80.7% and 61.1%, while in the imputed value, it was less than 15% (Appendix A).

These results suggest that false negativity correction effectively imputed the *P2ry12* expression values. Despite a significant decrease in the dropout ratio in the sham group of GES174574, the pattern of high *P2ry12* expression only in the microglia population remained unchanged.

### 2.2. Identification of MSPGs Using Integrative Analysis of scRNA-seq and Bulk-seq Data

To obtain MSPGs in the scRNA-seq data, we computed the Pearson correlation coefficient of each gene with *P2ry12* in microglia in different groups using the imputed data and integrated the coefficient in the same genes between GSE167593 and GES174574 using meta-analysis, based on the different groups. Of note, a correlation coefficient greater than 0.6 or less than −0.6 is considered to be moderately high, indicating a significant relationship between the variables. After filtering the correlation genes based on this criterion, 263 genes were found to be shared between the tMCAO and sham groups, including 262 positive correlation genes and 1 negative correlation gene (Appendix A). Further analysis of other populations using the same method showed that no genes other than *P2ry12* were found to be associated with *P2ry12*. These results suggested that *P2ry12*-associated genes were highly specific for microglia during IRI.

To improve the reliability and reproducibility of microglial *P2ry12*-associated gene expression, an integrative analysis of scRNA-seq and bulk-seq data was performed. Two published bulk-seq datasets of microglia sorted from the cerebral cortex in *mice* undergoing IRI were acquired, including five tMCAO and three sham in GSE77986 and two tMCAO and two sham in GSE172456. The Pearson correlation coefficient was calculated for each gene in relation to *P2ry12* in the different groups following the integration of bulk-seq data using Rank-in. To align with the correlation of microglia-specific *P2ry12* genes in scRNA-seq data, genes with a coefficient greater than 0.6 in both groups were considered as positive correlation genes, and those with a coefficient less than −0.6 in both groups were considered as negative correlation genes. In the bulk-seq data, 5062 genes were found to have a positive correlation and 5408 genes were found to have a negative correlation with *P2ry12*. The focus of this study was on the positive correlation genes in microglia as the only negatively correlated gene derived from the scRNA-seq data was not present in the negative correlation genes obtained from the bulk-seq data. A list of positive correlation genes was generated based on the ranking of correlation coefficients in the bulk-seq data. Furthermore, 40 sets of positive correlation genes in the scRNA-seq data were generated using consecutive correlation coefficient cut-off thresholds (ranging from 0.60 to 0.99 with increments of 0.01), and 41 different positive correlation rank lists were generated in the bulk-seq data using consecutive top percentage cut-off thresholds (ranging from 10% to 50% with increments of 1%). For identifying an optimal correlation coefficient to obtain MSPGs in the scRNA-seq data, we assessed the enrichment of each set of positively correlated genes in each rank list using the minimum hypergeometric test (Appendix A). We found that as the correlation coefficient cut-off threshold increased, the standard deviation of the corresponding enrichment *p*-values decreased and became stable (Appendix A).

In statistics, a strong positive correlation in statistics is defined as a correlation coefficient greater than 0.8 [25]. Further analysis of enrichment *p*-values within the range showed a stable standard deviation (Appendix A). Therefore, the standard deviation values within the range of correlation coefficient greater than 0.8 were utilized as a reference to calculate the Z-score for all standard deviation values in the whole range (a correlation coefficient greater than 0.6). The optimal correlation coefficient cut-off threshold was determined by the first index of a Z-score less than three, which indicates that 99.7% of the reference data were included in the final range. As a result, the optimal correlation coefficient was determined to be 0.74, leading to the identification of 88 MSPGs in the scRNA-seq data (Figure 2a).

### 2.3. MSPGs-Induced Scores Reproduce P2ry12 Expression in the scRNA-seq Data

To evaluate MSPGs expression in scRNA-seq data, we calculated the MSPGs-induced score for each cell using both the AUCell method and the AddModuleScore method (Figure 2b,c). Our results showed that microglial populations had substantially higher scores calculated by both methods, compared to other populations in the scRNA-seq data. This finding was consistent with the previously observed high expression levels of the *P2ry12* gene in scRNA-seq data, suggesting that the MSPGs-induced score is a reliable alternative to *P2ry12* gene expression in scRNA-seq data. To examine the changes in MSPGs-induced scores in the microglial population during IRI, we compared the scores between the sham and tMCAO groups using the Mann–Whitney U test. The results showed that the microglia in the sham group had significantly higher scores than those in the tMCAO group in the scRNA-seq data (*p*-value < 0.001). To determine the effect of the number of MSPGs on the overall scores, we randomly sampled 66 (75%) genes from the 88 MSPGs using the bootstrap method for 100 iterations and re-computed the scores independently. We observed a strong correlation between the re-computed scores and the original scores, with all correlation coefficients exceeding 0.97 (Figure 2d).

### 2.4. Decreased P2ry12 Expression in Murine Cerebral Cortical Samples during IRI

To investigate the ability of MSPGs to predict the change in *P2ry12* expression in the cerebral cortical samples of *mice* during IRI, we integrated four *mouse* bulk-seq datasets using Rank-in, including GSE28731, GSE58720, GSE32529, and GSE23163, consisting of 13 sham samples and 13 tMCAO samples. The average expression of MSPGs was calculated for each sample and compared between the two groups. The results showed that the average expression of MSPGs in the sham samples was significantly higher than that in the tMCAO samples (*t*-test *p*-value < 0.00001; Shapiro-Wilk test *p*-value > 0.05), as displayed in Figure 3a. To explore the scalability of MSPGs in rat cerebral cortical samples during IRI, we integrated three rat bulk-seq datasets associated with IRI using Rank-in, including GSE97537, GSE163614, and GSE199066, consisting of 8 sham samples and 10 tMCAO samples. The average expression of MSPGs in the sham samples was significantly higher than that in the tMCAO samples (*t*-test *p*-value < 0.00001; Shapiro–Wilk test *p*-value > 0.05), as shown in Figure 3b. To verify the findings, we employed qPCR and Western blot analysis to examine changes in *P2ry12* mRNA and protein expression in the ipsilateral cerebral cortex of rats subjected to tMCAO. Our results revealed a gradual decrease in *P2ry12* mRNA expression levels at three time points (1 h, 6 h, and 24 h) after tMCAO. Specifically, the 6 h and 24 h tMCAO groups showed significantly lower *P2ry12* mRNA levels compared to the sham group (Figure 3c). Additionally, there was a significant decrease in P2ry12 protein levels in the tMCAO group relative to the sham group. Furthermore, we evaluated *P2ry12* mRNA and protein expression in the penumbra of the cerebral cortex in tMCAO rats, and we observed a similar pattern of downregulation.

### 2.5. The Correlation between P2ry12 Expression and Microglial Activation

After analyzing the MSPGs-induced scores of the microglia in the tMCAO group using the AddModuleScore and AUCell methods, we observed a heterogeneous expression pattern in the scRNA-seq data (Appendix A). To explore the association of this heterogeneous expression pattern with microglial function, we used trajectory analysis to order the microglial cells extracted from the tMCAO group based on MSPGs expression and then projected these cells onto a reduced dimensional space constructed through principal component analysis (Figure 4a,b). The ordered microglial cells were divided into three states based on the expression levels of microglia-specific *P2ry12* genes: state 1 (high expression), state 2 (median expression), and state 3 (low expression). The performance of separation was further verified by plotting MSPGs-induced scores in the reduced dimensional space (Figure 4c,d and Appendix A). Differential gene analysis was then conducted for each state. We found that 291 differentially expressed genes were obtained in GSE167593 (Appendix A), with 183 significantly upregulated genes in state 1, 3 in state 2, and 105 in state 3. In GSE174574, 317 differentially expressed genes were obtained (Appendix A), with 168 significantly upregulated genes in state 1, 14 in state 2, and 135 in state 3. Subsequently, GO enrichment analysis was performed on the differentially expressed genes in different states. In GSE167593, the significantly upregulated genes in state 1 were enriched in processes related to cell development, protein folding, and the regulation of metabolic processes. Meanwhile, in state 3, the significantly upregulated genes were enriched in processes related to the response to stimulus and viral processes (Figure 4e). In GSE174574, the significantly upregulated genes in state 1 were enriched in antigen processing and presentation and cell development. For state 3, the significantly upregulated genes were enriched in processes related to the response to cell migration and the response to stimulus (Figure 4f). These findings indicate a correlation between low *P2ry12* expression and microglial activation during IRI.

### 2.6. The P2ry12-Mediated Cross-Talk between Microglia and Other Cortical Populations during IRI

To assess the effect of *P2ry12* expression on microglial interactions with other cortical populations during IRI, we conducted a cell–cell interaction analysis. The division of microglia into high-*P2ry12* and low-*P2ry12* expression groups was performed using microglial state 1 and state 3. We found that both low *P2ry12* and high *P2ry12* expression microglia exhibited upregulation of ligands, with 25 and 19, respectively. Furthermore, both groups showed upregulation of receptors, with 21 and 15, respectively. Further analysis showed that the set of upregulated ligands in low *P2ry12* expression microglia was a superset of those in high *P2ry12* expression microglia, and the same was true for the set of upregulated receptors (Appendix A). Our comparative analyses between low and high *P2ry12* expression microglia revealed 6 unique ligands and 6 unique receptors in low *P2ry12* expression microglia (Figure 5a). To enhance the reliability of these ligands and receptors, we validated them using integrated *mouse* microglia-sorted bulk-seq datasets (GSE77986 and GSE172456). Since low *P2ry12* expression in the MCAO group had been proven, we simply confirmed that the ligands and receptors retained higher expression levels in the MCAO group compared to the sham group, with the exception of *Tnfsf12*, which was not present in the bulk-seq datasets. Finally, three ligands (*Nampt*, *Cxcl2*, and *Igf1*) and one receptor (*Flt1*) were successfully validated (Figure 5b, Appendix A). The three ligands linked microglia to various cell types, including monocytes/macrophages, ependymal cells, neural progenitor cells, neurons, and granulocytes. The one receptor linked perivascular cells to microglia (Figure 5c).

### 2.7. Inference and Verification in P2ry12-Mediated Microglial Function

To further explore the functions of *P2ry12*-mediated microglia, a GO enrichment analysis was performed on MSPGs. We found significant enrichment (adjusted *p*-value < 0.05) for biological processes related to migration, such as positive regulation of macrophage migration and regulation of macrophage migration (Figure 6a). To examine the role of *P2ry12* in regulating microglial migration to vessels, an immunofluorescence assay was performed. In order to improve the accuracy of our immunofluorescence assessment, we initiated our study by conducting a histological examination using H&E and Nissl staining to determine the penumbra region within the ischemic lesion (Appendix A). Due to the fact that microglial proliferation in IRI affects the quantity of microglia attached to the vessel wall, we employed a corrected formula (detailed in the Methods section) to calculate the density of microglia attached to the walls of the vessels. The results indicated that there was a significant increase in the density of microglia adhered to vessel walls in the tMCAO group compared to the sham group (adjusted *p*-value < 0.05, Dunn’s test). However, treatment with PSB0739 in the tMCAO group resulted in a significant reduction in the density of microglia adhered to vessels compared to the tMCAO group alone (adjusted *p*-value < 0.05, Dunn’s test) (Figure 6b,c). These findings suggested that *P2ry12* played a role in regulating microglial migration to vessels.

## 3. Discussion

In the present study, we introduced a novel computational method based on the integrative analysis of scRNA-seq and bulk-seq data from the IRI-derived cerebral cortex in *mice*. This approach allowed us to identify gene sets significantly correlated to the gene *P2ry12* in microglia. The combined computational and experimental approach revealed a decrease in *P2ry12* expression during IRI and a correlation between the MSPGs and *P2ry12*. Integrative trajectory analysis and cell–cell interaction analysis further indicated that low *P2ry12* expression correlated with microglial activation. The GO enrichment analysis of MSPGs and in vivo experiments demonstrated that *P2ry12* regulates microglial migration toward blood vessels.

In previous studies, the expression of the microglial marker gene *P2ry12* has been reported in a diverse range of cerebral cortical cells [26,27]. This is consistent with scRNA-seq data, which show that *P2ry12* is expressed in multiple non-microglial populations. However, scRNA-seq data also reveal that *P2ry12* expression is significantly higher in microglia compared to other populations. This suggests that the total *P2ry12* gene expression in the cerebral cortex is largely contributed to by microglia and that the *P2ry12* mRNA expression levels in microglia can be considered representative of total *P2ry12* expression in the cerebral cortex.

Furthermore, gene sets correlated with *P2ry12* were only identified in microglia, not in other populations, in the scRNA-seq data (Appendix A). Additionally, a computational study and an in vivo experiment regarding the change in *P2ry12* expression in cortical microglia during IRI produced similar results, suggesting that MSPGs can be used as a suitable substitute for *P2ry12* in this context. These findings highlight the specificity of MSPGs for microglia during IRI, providing support for their potential use as a suitable substitute for *P2ry12* in this context.

Our results showing decreased expression levels of P2ry12 during IRI are in line with previous studies investigating CNS disorders such as Alzheimer’s disease [12,13], multiple sclerosis [14], and amyotrophic lateral sclerosis [28]. However, a discrepancy was found in the study by Villa et al., which reported increased expression of P2ry12 in microglia during ischemic injury [15]. This difference may be due to the different lesion areas and diseases studied. In our study, we focused on the comparison between microglia in the penumbra region and normal microglia, while Villa et al. compared microglia in the infarct region with those in the penumbra region. Additionally, it is important to note that while ischemic injury and IRI share some aspects of the injury process, they are recognized as having distinct pathogenic processes [29].

The results of our study revealed that microglia with low expression of *P2ry12* were more likely to be in an activated state compared to those with high *P2ry12* expression. This is in line with previous studies that have shown that more microglia become activated after 1–14 days of IRI [30]. Microglia activation can have both beneficial and harmful effects on the affected tissues, and the phenotype of activated microglia can significantly impact the outcome of injury [31]. Activated microglia are generally classified into two phenotypes: M1, the proinflammatory phenotype, and M2, the anti-inflammatory phenotype [32]. To further determine the phenotype of low-*P2ry12* microglia during IRI, we conducted cell–cell interaction analysis. Our findings showed that low-*P2ry12* microglia exhibited increased expression of three ligand-encoding genes, *Nampt*, *Cxcl2*, and *Igf1*, compared to high-*P2ry12* microglia. While *Nampt* and *Cxcl2* have been documented to have a proinflammatory effect on microglia, Igf1 has an anti-inflammatory effect [33,34,35]. This suggests that low-*P2ry12* microglia adopt a mixed M1/M2 phenotype, with a slight predominance of the M1 phenotype. Our results support the current understanding that microglial cells in different polarization states coexist and counteract each other during neuroinflammatory responses [36,37].

The cell–cell interaction analysis revealed a link between perivascular cells and low-*P2ry12* microglia through the ligand–receptor pair Pgf-Flt1. Pgf has been found to play a crucial role in the accumulation of microglia toward the lesion [38]. The findings suggest that low-*P2ry12* microglia have a higher potential for migration to blood vessels. This inference was indirectly supported by subsequent immunofluorescence assays, which showed that the density of microglia attached to blood vessels was significantly increased in the tMCAO group compared to the sham group. These results are consistent with previous studies on the dynamics of microglia association with blood vessels during ischemic injury [39]. Moreover, we observed a significant reduction in microglia adhesion to blood vessels in the PSB0739-treatment tMCAO group compared to the tMCAO group. Although both groups blocked the function of P2ry12, one by inhibiting protein activity and the other by inhibiting mRNA levels, this resulted in a distinct role of P2ry12 in microglial cells. The discrepancy may be attributed to the timing of P2ry12 inhibition, with the former gradually reducing P2ry12 levels post-IRI, while the latter inhibited it two hours prior to IRI. A previous study utilizing in vivo imaging found that microglia associated with blood vessels have a dual role in vessel function during neuroinflammation [40]. In the early stages, activated microglia aid in maintaining the integrity of the blood–brain barrier through interaction with endothelial cells. However, in later stages, they participate in the phagocytosis of vessels. Thus, we supposed that regulating microglial adhesion to vessels by inhibiting P2ry12 activity or expression at a suitable time point may alleviate IRI. Further research is necessary to comprehend the effects of inhibiting P2ry12 at different time points and to determine the optimal time window for administering inhibitors for maximum improvement of IRI.

Our study has several advantages. Firstly, imputing the technical zeros in raw scRNA-seq data is crucial for accurate downstream analyses. We employed the ALRA method, which effectively imputed the technical zeros and improved the power of correlation analyses. Secondly, the proposed computational method combines scRNA-seq and bulk-seq data to generate MSPGs, allowing for downstream analysis of both data types. Additionally, the MSPGs were used to investigate *P2ry12* expression in rats and were confirmed by subsequent in vivo experiments, demonstrating the applicability of MSPGs across species. One potential limitation of our study should be considered. While we were able to utilize an integrative analytic method to investigate the expression and function of microglial *P2ry12* during IRI, it is important to note that the high specificity of *P2ry12* for microglia may have played a critical role in the success of our approach. Therefore, it remains uncertain whether this approach could be equally effective in investigating genes that lack specificity for a particular cell type.

## 4. Materials and Methods

### 4.1. Data Acquisition

The data for this study were obtained from the Gene Expression Omnibus (GEO) database (https://www.ncbi.nlm.nih.gov/geo/) (accessed on 14 February 2023), based on the inclusion criteria of the availability of ipsilateral cerebral cortical samples from male rats and *mice* 24 h post tMCAO injury and sham control samples. To perform single-cell analyses, two scRNA-seq datasets (GSE167593 and GSE174574) were retrieved, which contained cerebral cortical samples of *mice* [41,42]. To determine the optimal cut-off threshold of MSPGs, two bulk-seq datasets (GSE77986 and GSE172456) were retrieved, which included microglia sorted from cerebral cortical samples of *mice* [43]. For the study of *P2ry12* expression, four bulk-seq datasets (GSE28731, GSE58720, GSE32529, and GSE23163) from cerebral cortical samples of *mice*, as well as three bulk-seq datasets (GSE97537, GSE163614, and GSE199066) from cerebral cortical samples of rats, were retrieved [44,45,46,47,48,49].

### 4.2. Processing of scRNA-Seq Data

For the purpose of quality control, the screening of cells and genes in each sample was carried out using three criteria. Firstly, genes expressed in at least three cells were selected for further analysis. Secondly, to minimize mitochondrial gene contamination, the cells were filtered based on mitochondrial gene counts, with a maximum proportion of 60% in GSE167593 and 50% in GSE174574. Finally, the cells with a total number of unique molecular identifiers per cell exceeding 6000 were excluded. To address the issue of technical dropout in the scRNA-Seq data, the ALRA algorithm was utilized for imputation. Data integration was performed on the PCA space using the Harmony algorithm (version 0.1.0) [50] and dimension reduction was performed using the RunTSNE function in the Seurat package.

### 4.3. Cell Type Annotation in scRNA-Seq Data

To identify an optimal automatic annotation method for the scRNA-seq data, we evaluated several commonly used automatic annotation tools and found that the combination of SingleR [51], scmap-cell [52], and SCINA [53] provided the best results (Appendix A). SingleR and scmap-cell are reference-based tools that utilized *mouse* cell type reference expression data from the celldex package. On the other hand, SCINA is a marker-based tool that annotates cell types based on brain cell type markers obtained from the CellMarker database (http://yikedaxue.slwshop.cn) (accessed on 14 February 2023) [54]. The final annotated label was assigned based on the most frequent label across the three tools.

### 4.4. Processing of Bulk-seq Data

The bulk chip sequencing data and the RNA sequencing data were normalized using the “normalizeBetweenArrays” function from the R-package limma [55] and the “estimateSizeFactors” function from the R-package DESeq2 [56], respectively. The normalized data were then transformed using a natural logarithm. Gene names were annotated to the probes using the R-package AnnoProbe and homologene conversion from *mouse* to *rat* was performed using the R-package homologene. The log-normalized bulk sequencing data were integrated using Rank-in [57], an online analysis tool for the integration of chip-seq and RNA-seq data (http://www.badd-cao.net/rank-in/index.html) (accessed on 14 February 2023).

### 4.5. Correlation Analysis among the Genes and the Minimum Hypergeometric Test

The Pearson correlation coefficient between *P2ry12* and each gene in the microglia population was analyzed from both scRNA-Seq data and integrated microglia sorted bulk-seq data. The integration of coefficients for the same genes across different scRNA-seq data was performed based on various groups using the metacor function in the R-package meta. The positive and negative correlation genes with *P2ry12* were ranked based on the absolute value of the sum of the positive and negative coefficients, respectively. To evaluate the enrichment of MSPGs in the rank list of *P2ry12*-associated genes from the integrated microglia sorted bulk-seq data, we performed the minimum hypergeometric test using the R-package mHG. The optimal correlation coefficient was determined by comparing the different enrichment *p*-values computed at various cut-off thresholds of the correlation coefficient.

### 4.6. Construction of Microglia-Specific P2ry12 Scores

The comprehensive scores of MSPGs in each cell were calculated using the R-package AUCell [58]. This package consisted of two main functions, AUCell_buildRankings and AUCell_calcAUC, with the former being utilized to rank all MSPGs in each cell according to gene expression values and the latter being utilized to calculate the fraction of the targeted gene set within the top 5% of the rankings. In addition, the comprehensive scores were also calculated using the AddModuleScore function implemented in the R package Seurat. Firstly, the calculation of comprehensive scores using the AddModuleScore function involved dividing all genes into 25 bins based on their average expression values across all cells. Secondly, the average expression value was determined by randomly sampling 100 genes from each bin, which served as an average control value. Finally, the expression values of each target gene in each cell were then subtracted by the average control value of the corresponding bin. The results were then averaged to obtain the microglia-specific *P2ry12* scores.

### 4.7. Single-Cell Trajectory Analysis and Differential Expression Genes Analysis

A single-cell trajectory analysis was performed using the Monocle R-package (version 2.22.0) [59]. Briefly, the MSPGs were passed to the function Monocle for ordering the microglial cells using the DDRTree algorithm. The microglial cells were divided into different states based on the segments of the tree. To determine the significantly upregulated genes in each state, a differential expression gene analysis was carried out utilizing the FindAllMarkers function in the Seurat R-package, with a logFC threshold greater than 0.5 and a *p*-value less than 0.01.

### 4.8. Cell–Cell Interaction Analysis

Cell–cell communication analysis was performed in the scRNA-Seq data using the R-package CellChat (version 1.4.0) [60]. The function identifyOverExpressedGenes was applied to compute over-expressed ligands or receptors in different populations based on the database of receptor–ligand interactions implemented in CellChat. The over-expressed ligand–receptor pairs were identified using the function identifyOverExpressedInteractions. The dropout of signaling genes was imputed by projecting the gene expression data onto the *mouse* protein–protein interaction network through the utilization of the function projectData. The probabilities of cell–cell communications were inferred using the function computeCommunProb with the default trimean method being employed to calculate the average gene expression per population.

### 4.9. Gene Ontology (GO) Enrichment Analyses

GO analysis was performed using the R-package clusterProfiler (version 4.2.2) [61]. A single gene set was subjected to the analysis utilizing the enrichGO function, while multiple gene sets underwent the analysis using the compareCluster function. A significance level for the false discovery rate was established at a value below 0.05 for determining significant GO function terms.

### 4.10. tMCAO Model and PSB0739 Dosing

In this study, adult male Sprague–Dawley *rats* were used to establish the tMCAO model as previously described [62]. They were housed in the Animal Experimental Center of Chongqing Medical University with a controlled temperature of 23 ± 2 °C and a 12/12 light cycle. Access to food and water was provided ad libitum. The *rats*, with a weight range of 250–280 g, were anesthetized using 3.5% chloral hydrate (350 mg/kg) and maintained at a body temperature of 37 ± 0.5 °C during the surgery with the use of a 37 °C thermostatic pad. The tMCAO model was established through the insertion of a nylon monofilament (Cinontech, Beijing, China) into the middle cerebral artery via the left external carotid artery, resulting in an ischemic status which was followed by reperfusion after removal of the monofilament after 1 h. During the surgery, local cerebral blood flow was monitored using a laser Doppler flowmeter (Periflux System 5000, Perimed, Sweden). The *rats* were sacrificed at 1 h, 6 h, and 24 h post reperfusion. Sham surgery was performed in a similar manner, but without the insertion of the monofilament.

The P2ry12 receptor antagonist, PSB0739 (GLPBIO, Montclair, CA, USA), was dissolved in physiological saline and freshly prepared on the day of use. Intra-cerebroventricular injection was carried out at a dose of 0.3 mg per kilogram, 2 h prior to tMCAO.

The *rats* studied were randomly assigned to the following groups: (1) the sham group, (2) the tMCAO group, and (3) the tMCAO group treated with PSB0739.

### 4.11. 2,3,5-Triphenyltetrazolium Chloride (TTC), Hematoxylin and Eosin (H&E) and Nissl Staining and Histological Evaluation

The *rats* were sacrificed at designated time points by administering an overdose of 3.5% chloral hydrate. The entire brain was immediately removed and frozen at −80 °C for 4 min. The brains were then sliced into five 2 mm-thick sections and stained with 2% 2,3,5-triphenyltetrazolium chloride (TTC, Sigma, St. Louis, MO, USA) at 37 °C for 20–30 min. The sections were subsequently fixed in 4% paraformaldehyde at 4 °C for 24 h and evaluated to confirm the success of the tMCAO model (Appendix A). For H&E and Nissl staining, the *rats* were sacrificed with an overdose of 3.5% chloral hydrate and transcardially perfused with 4% paraformaldehyde. Their brains were removed and fixed in paraformaldehyde for 24 h. The brains were then dehydrated and embedded in paraffin to produce 5 μm coronal brain sections. H&E and 1% toluidine blue dye (Nissl staining) were used to individually stain the sections using standard protocols. Neuropathological variations in all sections were evaluated using a light microscope.

H&E and Nissl staining were utilized to identify the location of the penumbra in an infarct. H&E staining was employed to highlight the global histologic variation in different regions (Appendix A). The ischemic core region was identified as pale and well demarcated, featuring substantial vacuolation and edema in the neuropils. Necrosis occurred in nearly all cells in this region, leading to significant accumulation of necrotic debris. Acidophilic cytoplasm with a pyknotic nucleus was observed in necrotic neurons. The penumbral region was characterized by selective neuronal necrosis, with a less severe degree of vacuolation and edema in the neuropil. Sparse necrotic neurons and some relatively normal neurons exhibited a peri-cellular halo appearance due to dendritic swelling. Furthermore, Nissl staining was employed to depict the variation of neuronal morphology in different regions (Appendix A). Normal neurons were identified by a large leptochromatic nucleus with a prominent nucleolus and Nissl substance extending into the dendrite but not the axon. In the penumbra, necrotic neurons were characterized by a shrunken nucleus with an inconspicuous nucleolus and Nissl substance that extended into both the dendrite and axon. In the ischemic core, necrotic neurons exhibited complete shrinkage of the nucleus with no visible nucleolus.

### 4.12. Double-Labeling Immunofluorescence Staining

Coronal brain sections of 5 μm thickness were obtained from different groups of brain tissue for immunofluorescence staining. Following deparaffinization and rehydration, heat-mediated antigen retrieval with citrate buffer (pH 6.0) was performed on the sections, and then they were blocked with 3% BSA. Overnight incubation at 4 °C was carried out with Iba1 (1:300, Abcam, ab178846, Cambridge, UK) and CD34 (1:100, ABclonal, A10796, Woburn, MA, USA). Subsequently, the sections were separately incubated with a 488-conjugated goat anti-rat IgG anitibody (1:200, Abbkine, green, A23220, Wuhan, China) and a cy3-conjugated goat anti-rat IgG anitibody (1:400, Abbkine, red, A22220, Wuhan, China) at 37 °C for 50 min. The sections, mounted with Vectashield-containing DAPI, were visualized using a fluorescence microscope.

The targeted areas for assessing the differences in microglia adherence to the vessel wall among the different groups were selected as the ischemic penumbra based on H&E and Nissl staining (Appendix A). Six high-power microscopic fields, with a field diameter of 0.5 mm and a magnification of 40×, were randomly captured within each group in the targeted areas. The number of microglia was determined by counting the DAPI-enveloped Iba1 staining, representing the number of microglial nuclei. The density of microglia adherence to the vessel wall was calculated by dividing the number of microglia adherent to the vessel wall by the number of vessels in each image. To counteract the rise in density caused by microglial proliferation during ischemic injury, instead of microglial migration to blood vessels, the corrected formula was as follows:$$d=\frac{M_{v}}{M_{t}\times V}$$
where d is the corrected density of microglia adherent to the vessel wall, M_v_ is the number of microglia adherent to the vessel wall, M_t_ is the total number of microglia, and V is the number of vessels.

### 4.13. RNA Isolation and Quantitative PCR (q-PCR)

To evaluate the difference in *P2ry12* gene expression among groups, q-PCR was carried out. Ipsilateral ischemic cerebral cortex or ischemic penumbral tissue samples were obtained, and total RNA was extracted using RNAiso Plus (TaKaRa Biotechnology, Dalian, China). cDNA was generated from the total RNA through reverse transcription with a PrimeScript RT Reagent Kit (TaKaRa Biotechnology). q-PCR was then performed on a CFX96 Touch™ Real-Time PCR Detection System using TaKaRa SYBR Premix Ex Taq II (Tli RNase H Plus) (TaKaRa Biotechnology) and *P2ry12* primer sequences (Sangon Biotech, Shanghai, China) with forward primer sequence 5′-TGGGCCTTCATGTTCCT-3′ and reverse primer sequence 5′-TGCCAGACCAGACCAAA-3′.

### 4.14. Western Blots

Western blot analysis was performed using protein extracted from the ipsilateral ischemic cerebral cortex or ischemic penumbral tissues. Equal amounts of protein (50 μg per lane) were loaded into SDS-PAGE gel wells for each sample. Following gel electrophoresis, the resolved proteins were transferred onto PVDF membranes (Millipore, Boston, MA, USA) through electrotransfer. The membranes were then incubated in a blocking solution (Tris-buffered saline containing 5% non-fat milk powder) for 2 h at room temperature and with a P2ry12 primary antibody solution (1:1000, 31255, Signalway Antibody, Greenbelt, MD, USA) overnight at 4 °C. The next day, the membranes were washed three times with 1X TBST for 10 min each. β-actin (1:5000, AC004, ABclonal) was used as a control. The protein bands were detected by an imaging densitometer (Bio-Rad, Hercules, CA, USA), and their gray values were measured using ImageJ/Fiji.

### 4.15. Statistical Analysis

All statistical analyses were conducted using R version 4.1.3 (https://www.r-project.org/) (accessed on 14 February 2023). Before conducting analysis on the continuous variables, the normality of the samples was assessed using the Shapiro–Wilk test. In the case of two independent samples with normality, a Student’s *t*-test was utilized for comparison, while a Mann–Whitney’s U test was employed if normality was not established. For multiple independent samples with normality, a one-way ANOVA was performed, followed by pairwise comparisons using Tukey’s test. If normality was not established for multiple independent samples or if the samples were not continuous variables, a Kruskal–Wallis test was conducted, followed by pairwise comparisons using Dunn’s Test. The level of statistical significance was set at *p* < 0.05.

## 5. Conclusions

Our study provides evidence that the expression of *P2ry12* in microglia decreases during IRI and that *P2ry12* plays a crucial role in controlling microglial migration toward blood vessels. These findings may inform future studies on the mechanisms of microglial migration and the optimal time for the administration of P2ry12 inhibitors. Furthermore, the novel computational approach we developed offers a promising tool for single gene expression and functional analysis in bioinformatics.

## Figures and Tables

**Figure 1 ijms-24-06772-f001:**
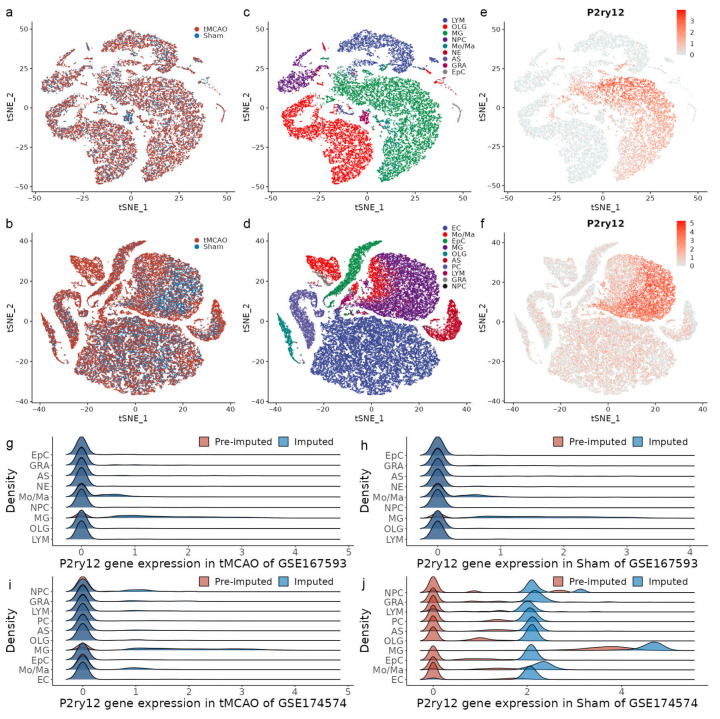
Distribution of *p2ry12* expression in single-cell transcriptomes of ischemia–reperfusion injury (IRI) in *mice*. (**a**,**b**) t-SNE visualization of single-cell transcriptomes from the transient middle cerebral artery occlusion (tMCAO)and sham control groups in GSE167593 and GSE174574, respectively. (**c**,**d**) t-SNE visualization of the transcriptomes of each cell population in GSE167593 and GSE174574, respectively. (**e**,**f**) t-SNE visualization of *P2ry12* expression in single-cell transcriptomes in GSE167593 and GSE174574, respectively. (**g**,**h**) Density plots of pre-imputed and imputed *P2ry12* expression in cell populations derived from the tMCAO and sham control groups in GSE167593, respectively. (**i**,**j**) Density plots of pre-imputed and imputed *P2ry12* expression in cell populations derived from the tMCAO and sham control groups in GSE174574, respectively. LYM, lymphocytes; OLG, oligodendrocytes; MG, microglia; NPC, neural progenitor cells; Mo/Ma, monocytes/macrophages; NC, neurons; AS, astrocytes; GRA, granulocytes; EpC, ependymal cells; EC, endothelial cells; PC, perivascular cells.

**Figure 2 ijms-24-06772-f002:**
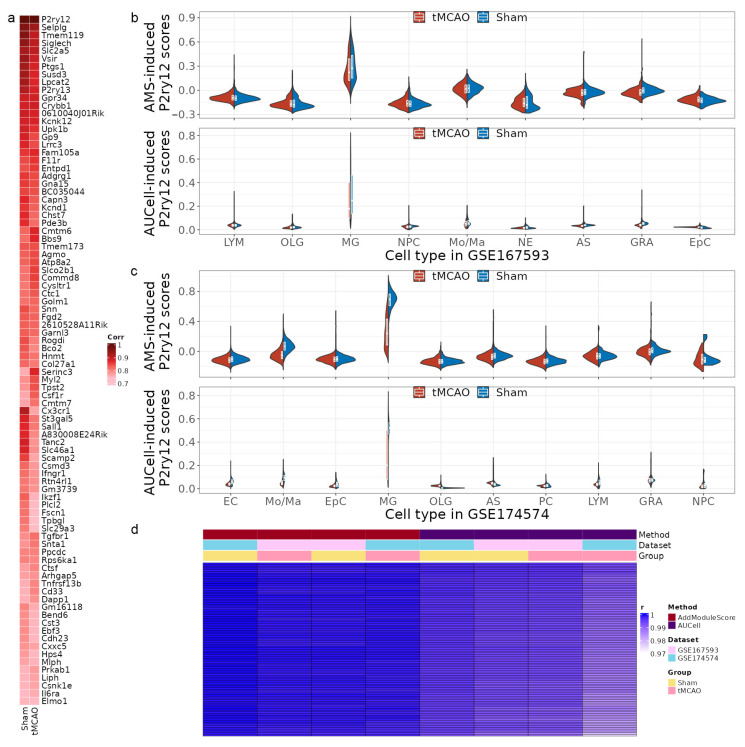
Distributions of scores induced by microglia-specific *p2ry12* genes (MSPGs) in different cell types in *mice*. (**a**) Correlation analysis between each gene in MSPGs and *P2ry12*. (**b**,**c**) The violin plots display the distribution of MSPGs-induced scores calculated by AddModuleScore and AUCell among different cell types in the GSE167593 and GSE174574 datasets, respectively. (**d**) Correlation heatmap demonstrating a strong correlation between the original MSPGs-induced scores and the recomputed scores using the bootstrap method. r, correlation coefficient; LYM, lymphocytes; OLG, oligodendrocytes; MG, microglia; NPC, neural progenitor cells; Mo/Ma, monocytes/macrophages; NC, neurons; AS, astrocytes; GRA, granulocytes; EpC, ependymal cells; EC, endothelial cells; PC, perivascular cells; AMS, AddModuleScore.

**Figure 3 ijms-24-06772-f003:**
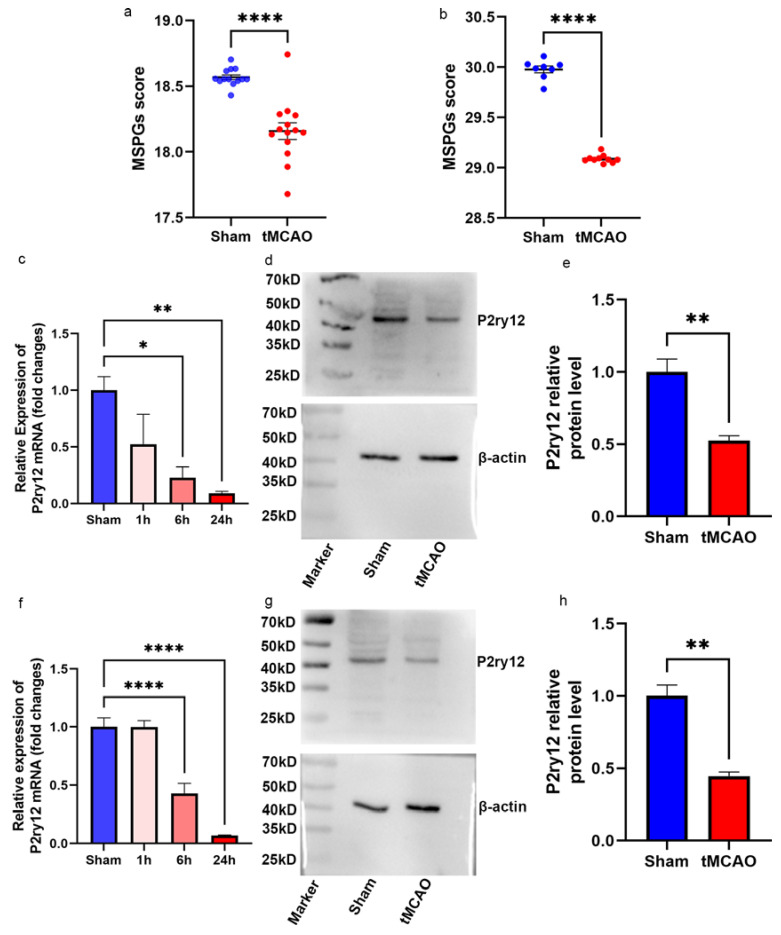
P2ry12 expression decreases during ischemia–reperfusion injury (IRI). (**a**) Average expression of microglia−specific *p2ry12* genes (MSPGs) in *mouse* datasets associated with IRI. (**b**) Average expression of MSPGs in rat datasets associated with IRI. (**c**) Quantitative PCR showing *P2ry12* mRNA expression in the normal rat cerebral cortex and ipsilateral ischemic rat cerebral cortex at three different post-reperfusion time points (1 h, 6 h, and 24 h). (**d**) Representative Western blot images depicting P2ry12 protein expression in the normal rat cerebral cortex and ipsilateral ischemic rat cerebral cortex at 24 h post reperfusion. (**e**) Quantification of the P2ry12 to β-actin ratio from the Western blot experiment. (**f**) Quantitative PCR showing *P2ry12* mRNA expression in the normal rat cerebral cortex and rat ischemic penumbra at three different post-reperfusion time points (1 h, 6 h, and 24 h). (**g**) Representative Western blot images depicting P2ry12 protein expression in the normal rat cerebral cortex and rat ischemic penumbra at 24 h post reperfusion. (**h**) Quantification of the P2ry12 to β-actin ratio from the Western blot experiment. The data are expressed as the mean ± SEM. * *p* < 0.05, ** *p* < 0.01, **** *p* < 0.0001.

**Figure 4 ijms-24-06772-f004:**
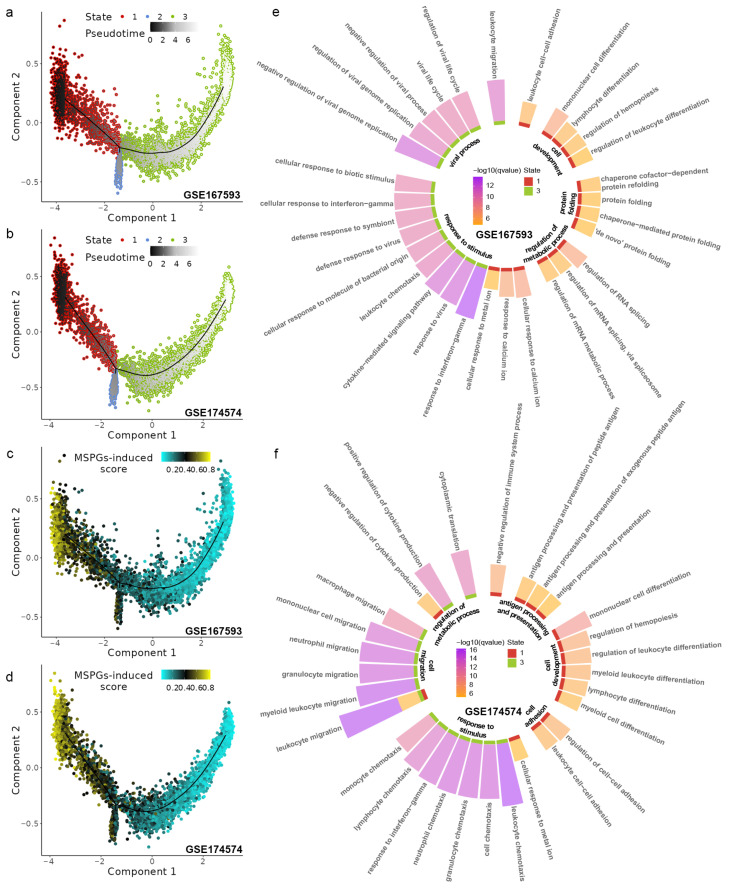
Exploring the effect of *p2ry12* expression on microglial function during ischemia–reperfusion injury (IRI) using single-cell data in *mice*. (**a**,**b**) Trajectory plots showing that microglial cells along the trajectory reconstructed by microglia−specific *p2ry12* genes (MSPGs) are divided into three states in GSE167593 and GSE174574, respectively. (**c**,**d**) Projection of MSPGs-induced scores calculated using AddModuleScore onto the trajectory in GSE167593 and GSE174574, respectively. (**e**,**f**) Circular bar plots showing the top 15 significantly enriched terms for microglia upregulated genes in each state in GSE167593 and GSE174574, respectively.

**Figure 5 ijms-24-06772-f005:**
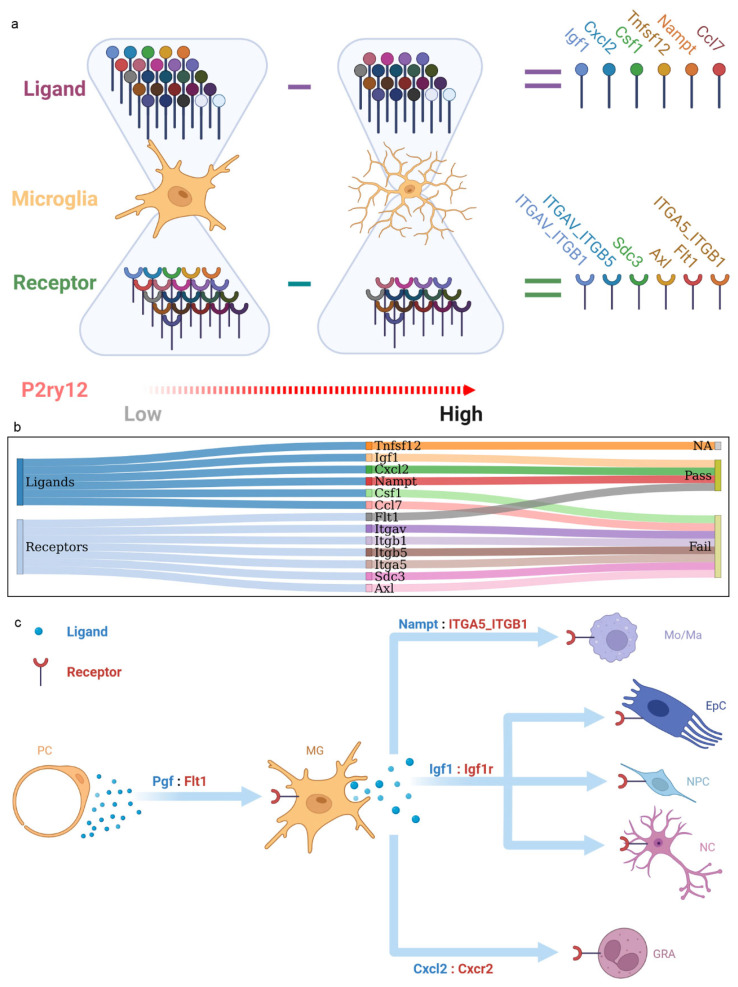
Interactions between *p2ry12* expression microglial populations and other cortical cell types. (**a**) Summary illustration of ligand- and receptor-encoding genes unique to low-*P2ry12* microglia compared to the high-*P2ry12* microglia in the single-cell data. (**b**) Sankey plot demonstrating the ligand- and receptor-encoding genes unique to low-*P2ry12* microglia that remain upregulated in the transient middle cerebral artery occlusion (tMCAO) group in bulk sequencing data. (**c**) Summary illustration of cell–cell interactions regarding the validated ligand–receptor pairs of low-*P2ry12* microglia. MG, microglia; NPC, neural progenitor cells; Mo/Ma, monocytes/macrophages; NC, neurons; GRA, granulocytes; EpC, ependymal cells; PC, perivascular cells. The figure was created with BioRender.

**Figure 6 ijms-24-06772-f006:**
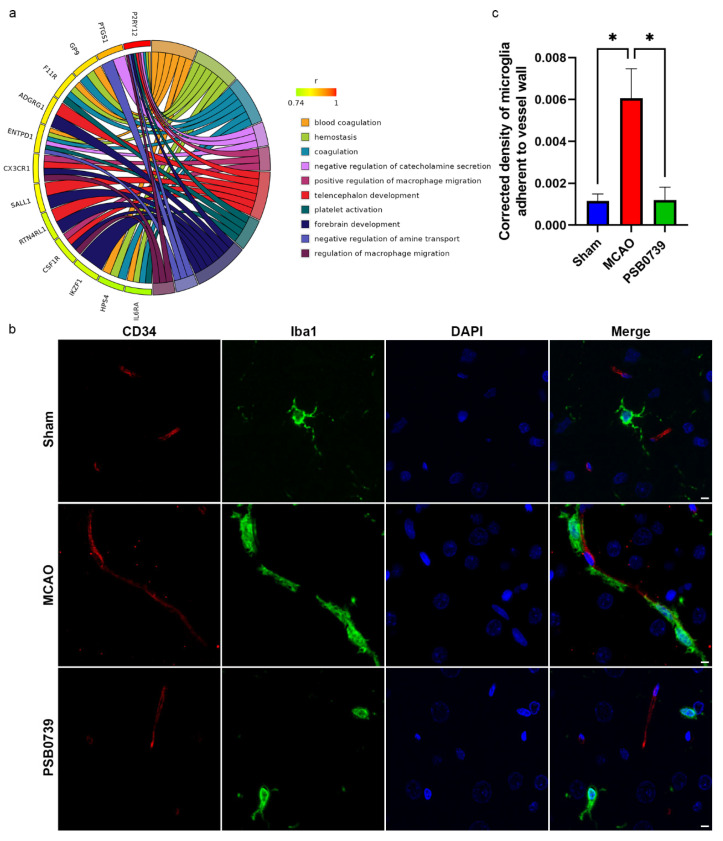
Effects of p2ry12 inhibition on microglial migration during ischemia–reperfusion injury (IRI). (**a**) The Gene Ontology enrichment analysis of microglia−specific *p2ry12* genes (MSPGs) for biological processes is shown in the chord plot. (**b**) The representative immunofluorescence images of Iba1 and CD34 (×400) illustrate the impact of P2ry12 antagonist PSB0739 on microglial adhesion to blood vessels during IRI compared to the sham group in rats. (**c**) The quantification of the vessel-adherent microglia to vessel ratio from the immunofluorescence staining indicates the effect of P2ry12 inhibition on microglial migration during IRI. The data are expressed as the mean ± SEM. * *p* < 0.05 (Dunn’s test). r, correlation coefficient. (Scale bars: 5 μm).

## Data Availability

The R script that implements the main workflow for the identification of correlated gene sets is available at: https://github.com/Patho-Lab/P2ry12 (accessed on 20 February 2023).

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
