# Peer review of "Integrative Analysis of Single-Cell and Bulk Sequencing Data Depicting the Expression and Function of P2ry12 in Microglia Post Ischemia–Reperfusion Injury"

_ijms, 2023, doi:10.3390/ijms24076772_

Round 1

Reviewer 1 Report

In this manuscript, Wang and colleagues employed computational methods and biochemical/molecular techniques using a rat model of transient middle cerebral artery occlusion (tMCAO) to examine the expression pattern of P2ry12 receptors in microglia. In their computational models, the authors focused on microglia-specific P2ry12 genes (or MSPGs) from data acquired from an ischemic reperfusion model. The authors found that the expression levels of P2ry12 receptors in microglia decreased over time, and these same observations were found with the computational model. In addition, the computational model revealed that there was an increase in expression of the Flt1 receptor gene, as well as increase in 3 ligand genes Nampt, Igf1 and Cxcl2). Overall, the manuscript is well written and organized. However, there are some concerns with the manuscript that need to be addressed:

1)    There is no discussion from the authors about the disadvantages with their approach, especially the computational approach. There is no mention of what assumptions may be incorrect. A few sentences on this would improve the manuscript.

2)    The western blot in figure 3d is not convincing at all. The authors MUST show the entire blot with a ladder to show the correct size of the proteins (P2ry12 and b-catenin). It is important to show the entire blot so that the reader can see how specific the antibody is and whether or not the antibody may or may not recognize other proteins.

3)    Why did the authors not perform western blotting assays on Flt1 expression levels? Demonstration that the expression levels of this proteins increased would significantly bolster their computational approach by verifying this observation.

4)    Minor: Some of the text in Figures 1, 2, 4 and 6 should be increased to make it legible.

Reviewer 2 Report

In this study, the authors investigated the expression and function of the P2ry12 gene in microglia during ischemia-reperfusion injury (IRI). The authors found that the expression of P2ry12 in microglia decreases during IRI, and that P2ry12 played a crucial role in controlling microglial migration toward blood vessels.

Comments

The reviewer has some concerns as follows:

1. One of major concerns is the methodology. The descriptions for these in vivo experiments are confusing and unconvincing. In the Methods section - 4.10. tMCAO model and PSB0739 dosing, the authors mention the details of the use of animals and IRI induction for the first time, but only mention the rats, not the mice. In 4.1. Data acquisition, the authors described that IRI mice were used. Are only rats used or both of rats and mice used for the analysis of Single-Cell and Bulk Sequencing Data? It should be carefully clarified the animal species used. Moreover, in 4.1. Data acquisition, it described the “ipsilateral cerebral cortical samples” from male mice were used; but in 4.11. Hematoxylin and eosin (HE) and Nissl staining and histological evaluation and 4.13. RNA isolation and quantitative PCR (q-PCR), the “ischemic penumbral tissue samples” were used. What are the real samples used in these experiments - whole ipsilateral cerebral cortical samples or penumbral tissue samples?

2. How can verify that the IRI model is successful? Some indicators can be shown.

3. In Figure 3, the P2ry12 expression during IRI in both mouse and rat was shown. In a and b, the animal species should be described clearly. In d, the immunoblot image for P2ry12 expression is not convincing that can be revised. Moreover, can P2ry12 receptor antagonist PSB0739 interfere with the P2ry12 protein expression?

4. In the legends of Figures 1, 2, 4, and 6, the animal species used should be clearly described.

Round 2

Reviewer 1 Report

The authors have adequately responded to my concerns and should be accepted for publication.

Reviewer 2 Report

This revised manuscript can be accepted. No further comments.